# A Novel Diagnostic Score Integrating Atrial Dimensions to Differentiate between the Athlete’s Heart and Arrhythmogenic Right Ventricular Cardiomyopathy

**DOI:** 10.3390/jcm10184094

**Published:** 2021-09-10

**Authors:** Valentina A. Rossi, David Niederseer, Justyna M. Sokolska, Boldizsar Kovacs, Sarah Costa, Alessio Gasperetti, Corinna Brunckhorst, Deniz Akdis, Felix C. Tanner, Firat Duru, Christian M. Schmied, Ardan M. Saguner

**Affiliations:** 1Department of Cardiology, University Heart Center Zurich, University Hospital Zurich, 8091 Zurich, Switzerland; valentina.rossi@usz.ch (V.A.R.); David.Niederseer@usz.ch (D.N.); justynamsokolska@gmail.com (J.M.S.); boldizsar.kovacs@usz.ch (B.K.); sarah.costa@usz.ch (S.C.); alessio.gasperetti@usz.ch (A.G.); Corinna.Brunckhorst@usz.ch (C.B.); deniz.akdis@usz.ch (D.A.); felix.tanner@usz.ch (F.C.T.); firat.duru@usz.ch (F.D.); Christian.Schmied@usz.ch (C.M.S.); 2Department of Cardiovascular Imaging, Institute of Heart Diseases, Wroclaw Medical University, 50-556 Wroclaw, Poland

**Keywords:** ARVC, sports medicine, atrial enlargement, echocardiography, task force criteria, right ventricle

## Abstract

Objective: The 2010 Task Force Criteria (TFC) have not been tested to differentiate ARVC from the athlete’s heart. Moreover, some criteria are not available (myocardial biopsy, genetic testing, morphology of ventricular tachycardia) or subject to interobserver variability (right ventricular regional wall motion abnormalities) in clinical practice. We hypothesized that atrial dimensions are useful and robust to differentiate between both entities and proposed a new diagnostic score based upon readily available parameters including echocardiographic atrial dimensions. Methods: In this observational study, 21 patients with definite ARVC were matched for age, gender and body mass index to 42 athletes. Based on ROC analysis, the following parameters were included in the score: indexed right/left atrial volumes ratio (RAVI/LAVI ratio), NT-proBNP, RVOT measurements (PLAX and PSAX BSA-corrected), tricuspid annular motion (TAM), precordial TWI and depolarization abnormalities according to TFC. Results: ARVC patients had a higher RAVI/LAVI ratio (1.76 ± 1.5 vs. 0.87 ± 0.2, *p* < 0.001), lower right ventricular function (fac: 29 ± 10.1 vs. 42.2 ± 5%, *p* < 0.001; TAM: 19.8 ± 5.4 vs. 23.8 ± 3.8 mm, *p* = 0.001) and higher serum NT-proBNP levels (345 ± 612 vs. 48 ± 57 ng/L, *p* < 0.001). Our score showed a good performance, which is comparable to the 2010 TFC using those parameters, which are available in routine clinical practice (AUC93%, *p* < 0.001 (95%CI 0.874–0.995) vs. AUC97%, *p* < 0.001 (95%CI 0.93–1.00). A score of 6/12 points yielded a specificity of 91% and an improved sensitivity of 67% for ARVC diagnosis as compared to a sensitivity of 41% for the abovementioned readily available 2010 TFC. Conclusions: ARVC patients present with significantly larger RA compared to athletes, resulting in a greater RAVI/LAVI ratio. Our novel diagnostic score includes readily available clinical parameters and has a high diagnostic accuracy to differentiate between ARVC and the athlete’s heart.

## 1. Introduction

The athlete’s heart is a physiological condition of morphological and functional alterations induced by repetitive and intense exercise. The correct identification and recognition of exercise-induced alterations allows the differentiation of healthy subjects from patients who suffer from an underlying structural heart disease potentially mimicking the athlete’s heart. Exercise-induced changes involving both ventricles have been widely analyzed [1].

There is a significant overlap between myocardial morphologic alterations described in arrhythmogenic right ventricular cardiomyopathy (ARVC) and the athlete’s heart, so that it is often challenging to distinguish the two conditions [2]. The 2010 Task Force Criteria (TFC) provide specific, although challenging, criteria to help distinguishing between both phenocopies [3]. The evaluation of RV wall-motion abnormalities is often subjective and affected by high inter-observer variability [4]. Furthermore, data from other diagnostic categories of the 2010 TFC, such as endomyocardial biopsy and genetic testing, are expensive, while the morphology of ventricular tachycardia is sometimes difficult to obtain.

Until now, observational echocardiographic studies reported contrasting findings for cardiac function and dimensions in athletes compared to ARVC [5,6,7]. Data about the physiological exercise-induced alterations involving the right atrium (RA) in athletes compared to patients with ARVC is scarce. In our clinical experience, we observed that in patients with ARVC—even at earlier disease stages—the RA is usually larger as compared to the left atrium (LA), whereas atrial dilation follows a symmetrical pattern in the athlete’s heart [8]. We thus hypothesized that atrial dimensional parameters and serum NT-proBNP levels help with differentiating between the athlete’s heart and ARVC.

The aim of our study was (1) to analyze dimensional changes of both atria in addition to conventional ventricular parameters in ARVC patients compared to a cohort of athletes, and (2) to provide a new diagnostic score developed upon readily available clinical parameters to help distinguish the athlete’s heart from ARVC in the clinical setting.

## 2. Methods

### 2.1. Population

Twenty-one consecutive patients with definite ARVC according to the 2010 TFC from the ARVC registry at the University Hospital of Zurich (www.arvc.ch (Accessed on 16 August 2021)) were matched 1:2 for age, gender and body mass index to 42 athletes referred for pre-participation screening. Athletes were defined as individuals engaging in regular exercise, training at amateur or professional levels based on a documented sports history [9]. Athletes were classified according to the level of intensity of dynamic or static exercise in endurance, mixed or power sports [10]. Demographics, clinical characteristics, transthoracic echocardiography (TTE), laboratory tests, sports and family history, medication, baseline 12-lead ECG and data from ergometry were collected from all patients.

The study was performed according to the Declaration of Helsinki and approved by the cantonal Zurich ethical committee (approval number KEK-ZH-NR:2014-0443). All patients prospectively included in our registry signed an informed consent prior to the study.

### 2.2. Echocardiography

A baseline TTE was performed at the time of first presentation and analyzed by experienced cardiologists. Left and right chamber measurements as well as RV outflow tract (RVOT) dimensions in the parasternal long axis (PLAX) and short axis (PSAX) were assessed according to current guidelines and adjusted for body surface area (BSA) [8,11]. Atrial volumes were measured at end-ventricular systole prior to the atrio-ventricular valve opening and corrected for BSA. The LA volume was calculated using the area-length method in the apical four- and two-chamber view, and the RA volume was measured using the area-length method in the modified apical four-chamber view [11] (Figure 1). The E/e’ (ratio between E-wave of mitral inflow as measure by pulsed-wave-doppler and e’, i.e. left ventricular early diastolic velocity as measured by tissue doppler) was measured by TTE in the apical four-chamber view as a measure of diastolic function with higher values indicating worse diastolic function. Further relevant measurements were performed accordingly [9]. Inter- and intra-observer variability was determined by re-examination of 20 randomly selected TTEs.

### 2.3. Ergometry

A baseline cycle ergometry with ramp protocol was performed at the time of first presentation. The maximal workload was calculated in Watts. Heart rate and blood pressure were measured at baseline and every minute during the test. A continuous ECG was registered. The double product factor (systolic blood pressure at peak of workload multiplied by the maximal pulse rate) was calculated.

### 2.4. Clinical Score to Differentiate the Athlete’s Heart from ARVC

Cut-off values for each variable were defined based on receiver-operating characteristic (ROC) curves. Quartiles were analyzed for parameters without normal distribution, and standard deviations were considered for parameters with normal distribution. Based on analysis of quartiles and standard deviations, 0 points were assigned for each parameter for which values were valid for >75% of athletes and for <25% of ARVC patients, 2 points were assigned for values valid for more than 75% of ARVC patients and for <25% of athletes, and 1 point was assigned for intermediate values. For dichotomic parameters (T-wave inversions (TWI) and depolarization abnormalities at baseline ECG), 1 point was assigned whenever this criterion was met. TWI were considered as pathological when present in at least two right precordial leads with extension beyond V3 in the presence of a complete right bundle branch block (RBBB). The same cut-offs values as in the 2010 TFC were considered for RVOT measurements. Sensitivity and specificity for ARVC diagnosis with the full 2010 TFC (all six categories, gold standard) were analyzed according to ROC curves. Our proposed novel score (Figure 2) ranges from a minimum of 0 points to a maximum of 12 points, with higher values suggesting a diagnosis of ARVC, and lower values suggesting a diagnosis of the athlete’s heart.

### 2.5. Statistical Analyses

All statistical analyses were performed with SPSS software (v25, SPSS Inc., Chicago, IL, USA). Continuous variables are presented as mean (± standard deviation) and categorical variables are expressed as percentages, unless stated otherwise. Differences in baseline characteristics between the groups were assessed by independent Student’s *t*-test, Mann-Whitney *U*-test, Pearson Chi-square test or Fisher’s Exact test, as appropriate. Univariable analysis for relevant clinical covariates was performed by Pearson or Spearman’s test, as appropriate. A two-sided *p*-value of <0.05 was considered to be statistically significant. Optimal cut-off values for the score calculation were calculated using ROC curves.

## 3. Results

### 3.1. Demographics and Classification of Sports Activity

Baseline characteristics of ARVC patients and athletes are summarized in Table 1. Twelve (57%) ARVC patients performed sports in the past and among these, *n* = 8 (67%) performed endurance sports, and *n* = 1 (8.3%) was a competitive endurance runner. All of them were advised to quit competition after being diagnosed with ARVC. Among the healthy athletes, *n* = 8 (19%) were professional, while the rest were competitive, non-professional athletes; *n* = 23 (54.8%) engaged in endurance sports and *n* = 19 (45.2%) in mixed activities.

ARVC patients were more likely to have a positive family history of sudden cardiac death or ventricular arrhythmia (*n* = 8 (38%) vs. *n* = 1 (2.4%), *p* < 0.001). On ergometry, both populations performed a maximal exercise test, as highlighted by the double product factor. Athletes performed better and reached a higher maximal heart rate compared to ARVC patients.

### 3.2. Echocardiographic Measurements

LA volumes were larger in athletes, whereas ARVC patients had larger RA volumes and RA areas and a higher RAVI/LAVI ratio compared to athletes (Figure 2).

RVOT, RV inflow tract and RV end-diastolic area were larger in ARVC patients compared to athletes. ARVC patients presented with lower parameters of RV systolic function. RV regional wall motion abnormalities were found in *n* = 10 (47.6%) ARVC patients and in *n* = 1 (2.4%) of athletes. Despite no significant differences in LV ejection fraction, ARVC patients had a higher proportion of LV wall motion abnormalities (*n* = 6, 28.6% vs. 0, *p* < 0.001). Similarly, ARVC patients had a worse LV diastolic function compared to athletes (E/e’ 8.1 ± 2.6 vs. 6.8 ± 1.6, *p* = 0.022; ARVC patients: 19% grade I diastolic dysfunction, 4.8% grade III vs. athletes; with 2.5% indeterminate diastolic function and the remaining athletes having a normal diastolic function, *p* = 0.011). A positive correlation between a higher RAVI/LAVI ratio and ARVC diagnosis, with a value of ≥1.11 being highly specific for ARVC, was found (*r* = 0.440, *p* < 0.001; AUC 82%, *p* < 0.001, for 1.11: sensitivity 63%, specificity 86%) (Table 2).

### 3.3. 2010 TFC and Novel Clinical Score

Details regarding the 2010 TFC are summarized in Table 3. Cut-off values for each variable used in our score are summarized in Table 4. The following best performing parameters were included: RAVI/LAVI ratio, NT-proBNP, PLAX RVOT/BSA, PSAX RVOT/BSA, TAM and electrocardiographic TWI in at least two precordial leads and depolarization abnormalities. ROC curves are shown in Figure 3. ROC curves for correct diagnosis of definite ARVC (gold standard 2010 TFC, all six categories) using our novel score showed an AUC of 93% (95%CI 0.874–0.995, *p* < 0.001). A score value obtained by our novel score of 6/12 yielded a specificity of 91% and a sensitivity of 67% for a diagnosis of definite ARVC, while a score of 7/12 yielded a specificity of 100% and a sensitivity of 57%.

ROC curves for the conventional 2010 TFC parameters (gold standard, all six categories) had the highest diagnostic performance (AUC 99%, *p* < 0.001, 95%CI 0.978–1.000). A sub-analysis of those 2010 TFC, which are easier to obtain (i.e., RVOT measurements or RV fractional area change (fac) and regional wall motion abnormalities on TTE as well as repolarization and depolarization abnormalities on 12-lead ECG) revealed an AUC of 97% for definite ARVC diagnosis (95%CI 0.93–1.00, *p* < 0.001), with a lower sensitivity of 41% compared to our novel score (67%).

### 3.4. Laboratory Testing

ARVC patients had higher serum NT-proBNP and C-reactive protein, while no significant differences were found in kidney function, leucocytes and their subpopulations, hemoglobin or TSH serum concentrations (Table 1).

### 3.5. Arrhythmia and ECG Alterations

Data are presented in Table 1. Two athletes (4.8%) were referred for further cardiologic investigations due to suspected arrhythmias during exercise, *n* = 2 (4.8%) for ECG anomalies at rest, and *n* = 1 (2.4%) due to pre-syncopal events. Among ARVC patients, *n* = 4 were referred due to syncope or presyncope, *n* = 1 for dyspnea, *n* = 5 for evaluation of ventricular arrhythmias, *n* = 1 for ECG alterations and *n* = 10 for further analyses due to known positive familiarity for ARVC. Compared to ARVC patients, athletes had less abnormalities on baseline 12-lead ECG (90.5 vs. 31%, *p* < 0.001), mainly TWI in the precordial leads beyond V2 in the absence of a complete RBBB (90.5 vs. 19%, *p* < 0.001). First degree atrio-ventricular block and RBBB were similar in both groups (*n* = 3 (14.3%) vs. 4 (9.5%), *p* = 0.571 and *n* = 5 (23.8%) vs. 4 (9.5%), *p* = 0.127, respectively).

## 4. Discussion

To the best of our knowledge, this is the first study assessing the diagnostic value of atrial dimensions and serum NT-proBNP to differentiate between ARVC and the athlete’s heart, integrating these findings into a novel diagnostic score to better discriminate between both phenocopies.

We report the following main findings:ARVC patients presented with significantly larger RA, but smaller LA dimensions as compared to athletes, resulting in a greater RAVI/LAVI ratio.The best novel diagnostic model to discriminate between ARVC (diagnosed by the full 2010 TFC serving as the diagnostic gold standard) and the athlete’s heart was obtained when including the following parameters: RAVI/LAVI ratio, RVOT in PLAX and PSAX, TAM, TWI and depolarization abnormalities on 12-lead ECG, and serum NT-proBNP with a diagnostic accuracy of 93%. A score value of ≥6 out of a maximum of 12 points was specific for a diagnosis of ARVC.Our novel diagnostic model showed a higher sensitivity to diagnose ARVC as compared to the 2010 TFC restricting to echocardiography and 12-lead ECG only, which are easy to obtain and readily available in daily clinical practice.

The findings of this study are clinically relevant as they help cardiologists and sport medicine specialists differentiate those subjects in daily clinical practice who are more likely to suffer from ARVC and thus may need further specific investigations.

### 4.1. Rationale to Develop a Novel Diagnostic Score to Differentiate between ARVC and the Athlete’s Heart

There is a significant overlap between myocardial RV morphologic alterations described in ARVC and the athlete’s heart [5,6,7]. The revised 2010 TFC were developed in order to improve the diagnosis of ARVC. However, the diagnostic cut-off values from these criteria were established based on a healthy, non-athletic control population, but not specifically designed to differentiate between ARVC and the athlete’s heart [3]. On the other hand, disease manifestation and progression in patients with ARVC can be exacerbated and accelerated by endurance sports [12].

Among the 2010 TFC criteria, the first one (RV imaging criterion) encompasses the evaluation of RV wall motion abnormalities and RVOT dilatation/RV dysfunction. This can be challenging since the echocardiographic window for the RV is narrow, particularly in dilated RVs, and lacks reproducibility [4,13]. Cardiac magnetic resonance imaging (CMR) is also not widely available, and assessment of RV regional wall motion abnormalities on CMR may also be challenging in certain scenarios [13]. False positive findings mimicking ARVC have frequently been reported in healthy probands [14]. Similarly, in the athlete’s heart the RV is dilated and often hypertrabeculated, thus making assessment of regional wall motion challenging. Moreover, athletes can develop ventricular arrhythmias from the RV related to a structural substrate, which has led to the hypothesis of an exercise-induced ARVC [15].

The second criterion of the 2010 TFC (tissue characterization by biopsy) is an invasive test usually not available in clinical routine, and often inconclusive due to the heterogeneous involvement of ventricular fibro-fatty infiltration, which spares the septum where biopsies are frequently taken from.

ECG depolarization and repolarization abnormalities (third and fourth category of the 2010 TC) are easy to obtain, however, a significant overlap between ARVC and the athlete’s heart may be present, and repolarization abnormalities may disappear during the course of disease in both phenocopies [8,16]. In this context, a relevant proportion of our athletic healthy population also presented with repolarization abnormalities. The fifth category (ventricular arrhythmias) is also often inconclusive since some patients do not present with ventricular tachycardia or ventricular tachycardia morphology has not been captured by 12-lead ECG. The sixth category (family history and genetic testing) can also be difficult to assess, given the fact that ARVC does not follow a familial pattern in up to 50% of cases, genetic test results are prone to misinterpretation, and desmosomal variants have also been identified in both a healthy population and athletes [17,18].

Per study design, the 2010 TFC criteria had the highest sensitivity and specificity for ARVC diagnosis (gold standard). In routine clinical practice, often only imaging and electrocardiographic criteria are available to help in the differential diagnosis between ARVC and the athlete’s heart. However, these criteria alone have a low sensitivity for a diagnosis of ARVC vs. athlete’s heart. On the contrary, the sensitivity of our novel diagnostic score outperformed these conventional parameters.

In our novel scoring system, we included well-established, readily available clinical and echocardiographic measurements that are less prone to interpretation with a very high specificity and good sensitivity with higher scores, making a diagnosis of ARVC more likely.

### 4.2. Cardiac Remodeling in the Athlete’s Heart and ARVC

The athlete’s heart is a physiological adaptation to repetitive and intense exercise leading to cardiac alterations. Data regarding the physiological exercise-induced alterations involving particularly the RA in athletes as compared to ARVC patients are scarce. Athletes undergo a repetitive right-sided volume overload which leads to a physiological enlargement of the right-sided cavities [1,19]. This finding is not associated with signs of increased right-sided pressures or markers of RV diastolic impairment and, thus, it is likely a physiologic remodeling [19].

Previous data indicates that the RA, but also LA, are commonly affected by the disease process in ARVC, where it is driven by pressure and volume overload secondary to RV diastolic and systolic dysfunction, although a primary desmosomal impairment at an atrial level has been suggested as well [20]. We report a disproportionate increase in RA dimensions in ARVC patients.

LA enlargement is another key component of the athlete’s heart, which is associated with a worse cardiovascular outcome [21]. The physiologic adaptation mechanisms behind LA enlargement are not clearly understood: It is likely directly related to a chronic LV volume and pressure overload or to chronic inflammation caused by excessive training [22]. LAVI is the most reliable and reproducible echocardiographic LA measurement [11]. In a recent meta-analysis, athletes had a LAVI 7 mL/m^2^ greater than the general population, although it is associated with preserved compliance and lower LA stiffness index [23,24]. We report accordingly greater LA volumes together with a significantly lower E/e’ ratio, indicating lower left-sided filling pressures in athletes compared to ARVC patients. We report a RAVI/LAVI ratio <1 in athletes (balanced and harmonic remodeling) and significantly higher (>1) in ARVC patients, where RA enlargement appreciably prevails. Therefore, we propose the RAVI/LAVI ratio as an objective, reliable and suitable screening imaging parameter to help distinguish between both phenocopies.

RV enlargement is a common clinical feature of the athlete’s heart [25]. Although the athlete’s heart is usually characterized by a normal systolic and diastolic function, a mild decrease in RV systolic function has been reported in athletes as well [26]. Accordingly, we report 41% of athletes with a fac ≤40%. Furthermore, 71% of athletes had a PLAX-RVOT, and 45% of athletes had a PSAX-RVOT diameter above the 2010 TFC cut-off values. Thus, we propose TAM, a robust and easy-to-assess echocardiographic parameter, as a further highly specific component to distinguish between ARVC and athlete’s heart.

### 4.3. NT-proBNP

NT-proBNP is an easily available blood test. Although in previous studies the suspicion for an acquired form of ARVC induced by strenuous physical activity was raised, athletes barely present NT-proBNP values exceeding 200 ng/L, while it is significantly higher in ARVC due to the underlying cardiomyopathy [27]. We found a positive correlation between higher NT-proBNP and ARVC diagnosis, with a value of ≥116 ng/L being highly specific for ARVC.

### 4.4. ECG Depolarization and Repolarization Alterations

According to current knowledge, TWI in precordial leads in adults correlate with RV structural anomalies and, as such, TWI beyond V2 in absence of a complete RBBB warrant further investigation [28]. In athletes, the majority of ECG alterations were TWI in at least two precordial leads including V2 or beyond, followed by incomplete RBBB. One athlete presented with a complete RBBB. As such, depolarization and repolarization anomalies represent a common finding in athletes [12].

### 4.5. Limitations

We performed our analysis based on TTE, since CMR data was not available in athletes, although it is considered the gold standard for the assessment of right-sided chambers and LA volumes. However, CMR is a time-consuming and expensive imaging modality, which is not used as a first-line screening method in clinical routine. Furthermore, this was a small observational study, and therefore the proposed clinical score needs validation in larger prospective cohorts.

## 5. Conclusions

ARVC patients present with significantly larger RA as compared to athletes, resulting in a greater RAVI/LAVI ratio. A novel diagnostic score encompassing those clinical parameters that are easy to obtain and robust (RAVI/LAVI ratio, RVOT dimensions in PSAX and PLAX, TAM, ECG repolarization and depolarization abnormalities and NT-proBNP) had a high diagnostic accuracy to differentiate between ARVC and the athlete’s heart.

## Figures and Tables

**Figure 1 jcm-10-04094-f001:**
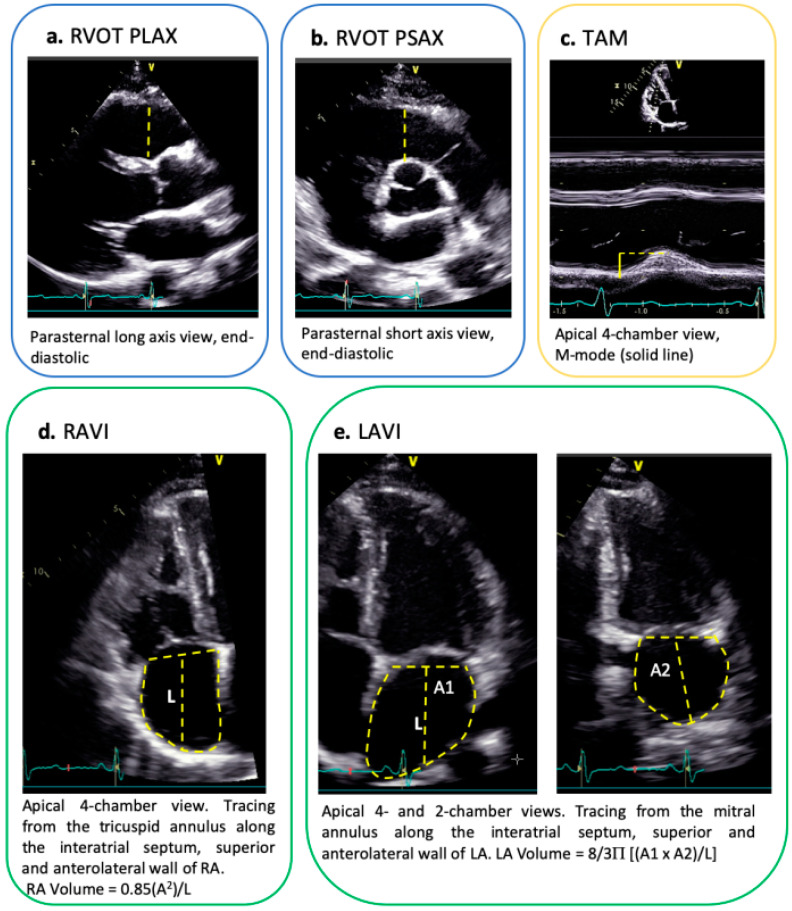
Echocardiographic measurements [11]. (**a**). RVOT measurement in parasternal long-axis view; (**b**). RVOT measurement in parasternal short-axis view; (**c**). TAM measurement from M-mode of four-chamber apical view; (**d**). apical four-chamber view with focus on the right side and 2D volumetric measurement of RAVI according to the area-length method; (**e**). apical four- (left) and two-chamber view with focus on the left atrium to measure the LAVI according to the area-length approximation. A1: atrial area in 4 chamber view, A2: atrial area in 2 chamber view. LAVI, left atrial volume indexed for body surface area; RAVI, right atrial volume indexed for body surface area; RVOT, right ventricular outflow tract; RV, right ventricle; TAM, tricuspidal annular motion.

**Figure 2 jcm-10-04094-f002:**
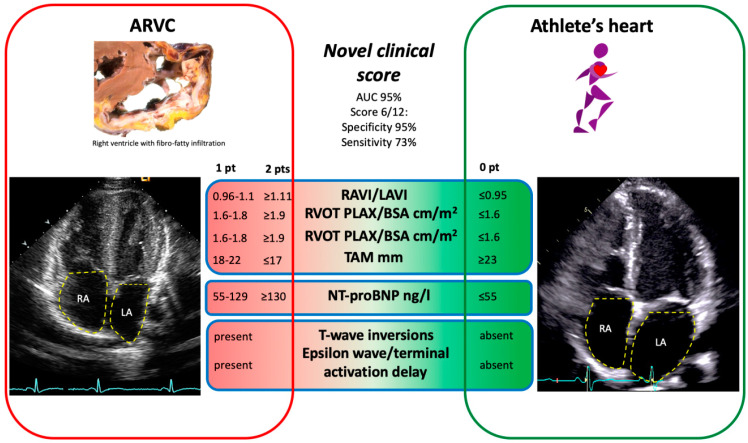
Novel score to differentiate the Athlete’s Heart from ARVC.

**Figure 3 jcm-10-04094-f003:**
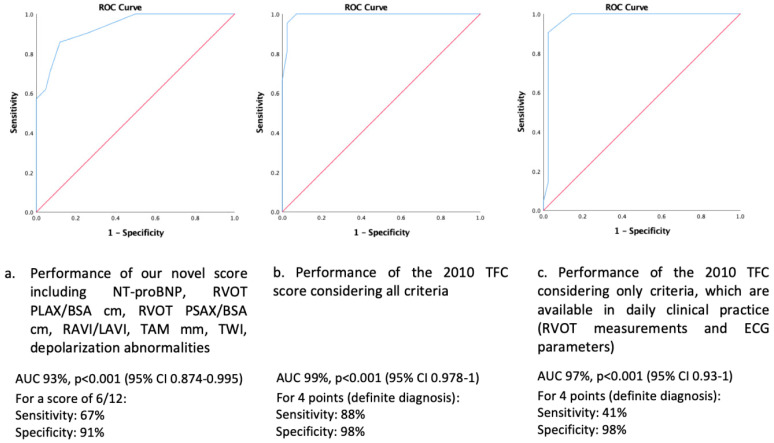
ROC curve for ARVC diagnosis. (**a**). performance of our novel score including NT-proBNP, RVOT PLAX/BSA, RVOT PSAX/BSA, RAVI/LAVI, TAM, TWI and depolarization abnormalities; (**b**). Performance of the 2010 Task Force Criteria score considering all criteria; (**c**). Performance of the 2010 Task Force Criteria score considering only criteria, which are available in daily clinical practice (RVOT measurements and ECG parameters). Abbreviations: NT-proBNP, N-terminal prohormone of brain natriuretic peptide; PLAX RVOT/BSA, right ventricular outflow tract measured from parasternal long axis indexed for body surface area; PSAX RVOT/BSA, right ventricular outflow tract measured from parasternal short axis indexed for body surface area; RAVI/LAVI, ratio between right atrial and left atrial volume indexed for BSA; TAM, tricuspid annular motion velocity; TFC, Task Force Criteria; TWI, T-wave inversion.

**Table 1 jcm-10-04094-t001:** Baseline characteristics of ARVC patients vs. athletes.

	ARVC *n* = 21	Athletes *n* = 42	
	Mean (SD) or %	Mean (SD) or %	*p*-Value
Gender (male/female)	16/5	33/9	0.830
Age (years)	37.3 (17.6)	32.4 (12.5)	0.204
BMI (kg/m^2^)	22.7 (5.9)	22.7 (2.5)	0.992
Sports %	57	100	
endurance/mixed/strength (*n*)	8/4/0	23/19/0	0.462
Among these: competitive athletes or professional (*n*)	1	8	0.417
Co-morbidities/medication			
Atrial fibrillation/flutter %	4.8	2.4	0.624
Arterial hypertension %	4.8	11.9	0.349
Beta-blocker	61.9	4.8	<0.001
Amiodarone	14.3	0	0.013
Diuretics	9.5	0	0.045
Aldosterone-Antagonists	14.3	0	0.013
ACE-I/Sartans	23.8	9.5	0.137
ECG at baseline			
HR at rest (bpm)	56.9 (8.9)	55.8 (11.5)	0.691
AV-block 1st degree %	14.3	9.5	0.571
T-wave inversions beyond V2 %	90.5	19	<0.001
Right bundle branch block %	23.8	9.5	0.127
24 h Holter ECG	*n* = 21	*n* = 24	
SVPB count/24 h	768 (1013)	768 (1550)	0.999
PVC count/24 h	2331 (2812)	111 (219)	<0.001
Blood test			
Leucocyte total G/L	6.7 (2.3)	5.9 (1.7)	0.194
Hb g/dl	144.6 (7.3)	143.8 (13.6)	0.806
NT-proBNP ng/L	345 (612)	48 (57)	<0.001
CRP mg/L	2 (2.6)	0.8 (0.6)	0.021
Cholesterin total mmol/L	4.3 (0.7)	5.4 (5.7)	0.518
GFR (CKD-EPI, mL/min)	94.1 (21.9)	100.3 (18.6)	0.296
TSH mU/L	1.6 (1)	2.1 (1)	0.152
Ergometry	*n* = 13	*n* = 33	
Maximum Watt	179.8 (75.3)	301 (86.8)	<0.001
% of expected maximum Watt	109 (35.6)	161.5 (36.5)	<0.001
HR max (bpm)	149.1 (30.5)	171.2 (23.1)	0.007
% of expected HR max	85.1 (14.7)	95.6 (12.8)	0.017
Double product factor	3.3 (0.9)	3.6 (0.8)	0.262
BP syst max mmHg	181.4 (30.8)	205.2 (25)	0.007
BP diast max mmHg	82.5 (11.3)	88.8 (12.5)	0.099
ECG alteration/arrhythmia %	61.5	9.1	<0.001

Abbreviations: ACE-I, ACE-inhibitors; AV, atrioventricular; BMI, body mass index; BP, blood pressure; bpm, beats per minute; CRP, C-reactive protein; GFR, glomerular filtration rate; Hb, hemoglobin; HR, heart rate; PVC, premature ventricular beats; SVPB, supra-ventricular premature beats; TWI, T-wave inversion >V2.

**Table 2 jcm-10-04094-t002:** Echocardiographic characteristics and 2010 Task Force Criteria for ARVC diagnosis.

	ARVC Patients *n* = 21 Mean (SD) or %	Athletes *n* = 42 Mean (SD) or %	*p*-Value
Echocardiography			
LAVI mL/m^2^	27.2 (17.7)	33.4 (7.8)	0.059
RAVI mL/m^2^	40.7 (29.8)	28.5 (8.7)	0.017
RAVI/LAVI ratio	1.76 (1.5)	0.87 (0.2)	<0.001
RA/LA axis ratio	1.2 (0.4)	0.97 (0.1)	0.004
PLAX RVOT cm	3.6 (0.7)	3.1 (0.5)	0.002
PLAX RVOT/BSA cm/m^2^	2.2 (1.2)	1.7 (0.3)	0.013
PSAX RVOT cm	3.7 (0.8)	3 (0.5)	<0.001
PSAX RVOT/BSA cm/m^2^	2.2 (1.2)	1.6 (0.3)	0.003
RVIT cm	4.4 (0.9)	3.8 (0.4)	0.001
RVIT/BSA cm/m^2^	2.5 (1)	2 (0.3)	0.003
RV ED short axis cm	4.2 (0.9)	4.1 (0.5)	0.478
RV Area D cm^2^	31.2 (8.7)	24 (4.1)	<0.001
fac%	29 (10.1)	42.2 (5)	<0.001
TAM mm	19.8 (5.4)	23.8 (3.8)	0.001
LV ED short axis cm	4.9 (0.6)	5.2 (0.6)	0.078
RV/LV ratio	0.9 (0.2)	0.8 (0.1)	0.029
LV shortening %	32.7 (10.5)	35.7 (6.2)	0.155
LV posterior wall mm	0.8 (0.1)	0.8 (0.1)	0.171
LV septal wall mm	0.9 (0.2)	0.9 (0.1)	0.575
LVEF %	55.8 (11.4)	56.9 (9.2)	0.683
LVEDVI mL/m^2^	60.8 (12.5)	73.7 (14.2)	0.001
LVMNI g/m^2^	79.5 (32.2)	83.6 (24.2)	0.580
rTh	0.3 (0.1)	0.3 (0.1)	0.893
LV wall motion anomalies %	33.3	0	<0.001
Diastolic dysfunction %	23.8	0	<0.001
E/e’	8.1 (2.6)	6.8 (1.6)	0.022

Abbreviations: BSA, body surface area; E/e’, ratio between E-wave of mitral inflow as measure by pulsed-wave-doppler and e’, i.e. left ventricular early diastolic velocity as measured by tissue doppler; ED, end-diastolic; Fac, fractional area change; LA, left atrium; LAVI, left atrial volume indexed for BSA LAVI/RAVI, ratio between left atrial and right atrial volume indexed for BSA; LA, left atrium; LAVI, left atrial volume indexed for BSA; LV, left ventricular; LVEDVI, LV end-diastolic volume indexed for BSA; LVEF, LV ejection fraction; LVMNI, LV mass index; NT-proBNP, N-terminal prohormone of brain natriuretic peptide; RA, right atrium; RAVI, right atrial volume indexed for BSA; rTh, relative wall thickness; RVIT, RV inflow tract; RVOT PLAX, right ventricular outflow tract measured from parasternal long axis; RVOT PSAX, right ventricular outflow tract measured from parasternal short axis; TAM, tricuspid annular motion.

**Table 3 jcm-10-04094-t003:** 2010 TFC and genetic analyses in ARVC patients and athletes.

Criterion	ARVC Patients	Athletes
I. Global or regional dysfunction and structural alterations on TTE		
Major (*n*)	20	1
Minor (*n*)	1	
II. Tissue characterization of RV wall*		
Major (*n*)	3	-
Minor (*n*)	1	-
Biopsy available in *n* = 5 ARVC patients and none among athletes		
III. Repolarization abnormalities		
Major (*n*)	11	6
Minor (*n*)	6	1
IV. Depolarization abnormalities		
Major (*n*)	3	0
Minor (*n*)	4	1
V. Ventricular Arrhythmias		
Major (*n*)	3	1
Minor (*n*)	13	1
VI. Family history		
Major (*n*)	10	0
Minor (*n*)		1
Pathogenic/likely pathogenic genetic variants	*n* = 16	
PKP-2 *n* = 8 DSG-2 *n* = 4 DSP *n* = 1 DSC-2 *n* = 1 SCN5A *n* = 1 TTN *n* = 1		

Abbreviations: AV, atrio-ventricular; DSC-2, desmocollin-2; DSG-2, desmoglein-2; DSP, desmoplakin; Fac, fractional area change; PKP-2, plakophilin-2; TTN, titin; PVC, premature ventricular beats; RVOT PLAX, right ventricular outflow tract measured from parasternal long axis by TTE; RVOT PSAX, right ventricular outflow tract measured from parasternal short axis by TTE; TWI, T-wave inversion.

**Table 4 jcm-10-04094-t004:** Clinical score.

	Value	Sensitivity, %	Specificity, %	*p*-Value	0 Points	1 Point	2 Points
NT-proBNP ng/L	116	52	91	<0.001	≤35	36–115	≥116
RVOT PLAX/BSA cm/m^2^	1.9	43	80	0.004	≤1.6	1.6–1.8	≥1.9
RVOT PSAX/BSA cm/m^2^	2.1	33	93	0.001	≤1.8	1.8–2.0	≥2.1
RAVI/LAVI	1.11	63	91	<0.001	≤0.95	0.96–1.1	≥1.11
TAM mm	17	77	99	0.003	≥23	18–22	≤17
TWI *				<0.001	absent	present	
Depolarization abnormalities **				<0.001	absent	present	

Sensitivity and specificity were calculated with ROC curves; *p*-values were derived from Pearson (for continue variables) or Spearman (for dichotomic variables) correlations between each parameter and ARVC diagnosis; * TWI were considered as pathologic when negative beyond V2 in absence of a complete right bundle branch block according to the 2010 TFC; ** depolarization abnormalities were considered as pathologic according to 2010 TFC. Abbreviations: NT-proBNP, N-terminal prohormone of brain natriuretic peptide; PLAX RVOT/BSA, right ventricular outflow tract measured from parasternal long axis indexed for body surface area; PSAX RVOT/BSA, right ventricular outflow tract measured from parasternal short axis indexed for body surface area; RAVI/LAVI, ratio between right atrial and left atrial volume indexed for BSA; TAM, tricuspid annular motion velocity; TFC, Task Force Criteria; TWI, T-wave inversion.

## Data Availability

Data are available from the corresponding author upon reasonable request.

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
