# Peer review of "A Novel Diagnostic Score Integrating Atrial Dimensions to Differentiate between the Athlete’s Heart and Arrhythmogenic Right Ventricular Cardiomyopathy"

_jcm, 2021, doi:10.3390/jcm10184094_

Round 1
Reviewer 1 Report
The authors aimed to assess whether simple and easily obtained parameters help in differentiation of the ARVC and the athlete’s heart. They included 21 patients with definite (according to 2010 Task Force Criteria) diagnosis of ARVC and 42 matched controls. They concluded that right atrium is significantly larger in patients with ARVC. Moreover, the authors developed a novel diagnostic score integrating atrial dimensions to differentiate between the athlete’s heart and ARVC.
The study is well designed, written, and presented. The findings of the study are clinically relevant.
I have only some minor remarks:
- TAM means tricuspid annular motion, not tricuspid annular motion velocity as it is written in the abstract (line 23).
- I do not agree with the statement that it is expensive to assess morphology of VT.
- I would change the place where Figure 1 is mentioned. It is mentioned in line 169 – please consider moving it to line 112.
- In table 3 ventricular arrhythmias are marked with two asterisks while it should be marked depolarization abnormalities.
- There is a typo error in the legend for figure 2 (it is written ROS instead of ROC).
Author Response
Reviewer 1
The authors aimed to assess whether simple and easily obtained parameters help in differentiation of the ARVC and the athlete’s heart. They included 21 patients with definite (according to 2010 Task Force Criteria) diagnosis of ARVC and 42 matched controls. They concluded that right atrium is significantly larger in patients with ARVC. Moreover, the authors developed a novel diagnostic score integrating atrial dimensions to differentiate between the athlete’s heart and ARVC.
The study is well designed, written, and presented. The findings of the study are clinically relevant.
Thank you for your appreciation.
I have only some minor remarks:
1. TAM means tricuspid annular motion, not tricuspid annular motion velocity as it is written in the abstract (line 23).
Answer to point 1: Thank you for noticing it. We have now corrected it in the abstract session.
2. I do not agree with the statement that it is expensive to assess morphology of VT.
Answer to point 2: The adjective “expensive” is mainly referred to the performance of endomyocardial biopsy and genetic testing. We have now rewrote it (line 76-77): “Furthermore, data from other diagnostic categories of the 2010 TFC such as endomyocardial biopsy, and genetic testing are expensive, while morphology of ventricular tachycardia is difficult to obtain.”
3. I would change the place where Figure 1 is mentioned. It is mentioned in line 169 – please consider moving it to line 112.
Answer to point 3: We have now added it to line 112, as well.
4. In table 3 ventricular arrhythmias are marked with two asterisks while it should be marked depolarization abnormalities.
Answer to point 4: Thank you for noticing it. As it was a typo initially referred to the number of patients, we deleted it now.
5. There is a typo error in the legend for figure 2 (it is written ROS instead of ROC).
Answer to point 5: Thank you for noticing it. We have now corrected it.
Reviewer 2 Report
In this paper, the authors aimed to establish a diagnostic score, based mainly on standard echocardiographic measurements, for the differentiation between athlete's heart and arrhythmogenic cardiomyopathy (ARVC). The study showed that patients with ARVC have larger right atria and right/left volumes ratio, also the proposed diagnostic score showed good sensitivity and specificity compared to standard criteria for ARVC.
It is an interesting study, with practical implications by suggesting a simplified approach for the initial differentiation of two conditions that require specific management.
The study protocol is adequate, the paper is well-written with a logical flow of data. I do not have specific concerns regarding the paper.
Author Response
Thank you for your positive evaluation of our paper. We have performed a spell check of the whole paper.